# WebCode2M: A Real-World Dataset for Code Generation from Webpage Designs

## Abstract

Automatically generating webpage code from webpage designs can significantly reduce the workload of front-end developers, and recent *Multimodal Large Language Models* (MLLMs) have shown promising potential in this area. However, our investigation reveals that most existing MLLMs are constrained by the absence of high-quality, large-scale, real-word datasets, resulting in inadequate performance in automated webpage code generation. To fill this gap, this paper introduces WebCode2M, a new dataset comprising 2.56 million instances, each containing a design image along with the corresponding webpage code and layout details. Sourced from real-world web resources, WebCode2M offers a rich and valuable dataset for webpage code generation across a variety of user scenarios. The dataset quality is ensured by a highly accurate scoring model that filters out instances with aesthetic deficiencies or other incomplete elements. To validate the effectiveness of our proposed dataset, we introduce a baseline model based on the *Vision Transformer* (ViT), named WebCoder, and establish a benchmark for fair comparison. Additionally, we introduce a new metric, TreeBLEU, to measure the structural hierarchy recall. The benchmarking results demonstrate that our dataset significantly improves the ability of MLLMs to generate code from webpage designs, confirming its effectiveness and usability for future applications in front-end design tools. Finally, we highlight several practical challenges introduced by our dataset, calling for further research. We have hosted the WebCode2M on an anonymous webpage: https://webcode2m-anonymous.github.io.

## 1 Introduction

Front-end software developers typically create webpages based on *Graphical User Interface* (GUI) mockups designed by UI designers. However, this process is often time-consuming and costly. To this end, several neural models have been proposed to automate the process of generating code from GUI design images, thereby alleviating the burden on front-end developers. Among these, pix2code [14] and sketch2code [48] are two exemplary works that translate images, whether simple-styled UI designs or hand-drawn sketches, into front-end code. Recently, *Multimodal Large Language Models* (MLLMs), such as GPT-4V [42], have also demonstrated impressive potential in this area.

Despite its potential, we are still far from fully automating front-end engineering to achieve true "screenshot in, code out" functionality. In particular, as highlighted in a recent work [53], the complexity of code generation increases with the increase in the total number of *HyperText Markup Language* (HTML) tags, the diversity of unique tags, and the depth of the Document Object Model (DOM) tree. Proprietary MLLMs, such as GPT-4V, also exhibit a notable decline in performance when confronted with real-world webpage designs that feature complex structures and a larger variety of unique HTML tags [53].

One possible solution lies in fine-tuning pre-trained LLMs, with the potential for improved performance as the amount of data increases. However, this approach faces a significant limitation because existing datasets are either too small to provide meaningful generalization [21, 53] or consist of synthetic data that does not fully capture the complexity and variability of real-world webpage designs [28, 62]. For instance, Design2Code [53] contains only 484 real-world samples, intended solely for testing and insufficient for effective fine-tuning. WebSight [28] is another dataset comprising approximately 0.8 million synthesized samples generated by LLMs. However, a significant disparity exists between these samples and real-world data [53]. Specifically, WebSight samples average 647 tokens, 19 tags, and a DOM depth of 5, whereas our study reveals that real-world samples can involve up to 50 times more tokens, six times as many tags, and double the DOM depth (See Fig. 2). This substantial gap between synthetic and real-world data can limit the practical effectiveness of fine-tuned MLLMs when applied to more complex, real-world scenarios.

**Our Work.** To fill this gap, this paper introduces a large-scale real-world dataset for webpage generation, named WebCode2M, which includes 2.56 million instances. Each instance features a high-quality webpage design image paired with its corresponding HTML and *Cascading Style Sheets* (CSS) code. This dataset overcomes the limitations of existing datasets by offering a diverse and comprehensive collection of real-world webpage designs and their associated code. On average, the samples contain 31,216 tokens, 158 tags, and a DOM depth of 13. WebCode2M is poised to be an invaluable resource for advancing the development of webpage code generation models.

To construct our dataset, we first collect approximately 0.5 billion real-world webpages from the Common Crawl dataset [2], which includes a diverse array of web domains and styles. For each webpage, we extract the associated CSS code and image elements, remove noise and irrelevant code, and generate screenshots. To ensure data quality, we develop a scoring model to filter out instances with incomplete elements or suboptimal aesthetic quality, such as disorganized layouts or excessive blank spaces, as illustrated in Figure 9 (See Appendix). This scoring model is trained on a manually annotated subset of 10,000 entries, curated by six annotators using consensus-based annotation, achieving a validation accuracy of 90% in distinguishing high- from low-quality instances.

To demonstrate the potential of our dataset for improving automatic webpage generation, we fine-tune a ViT model [17] as a new baseline for translating webpage design images into HTML/CSS

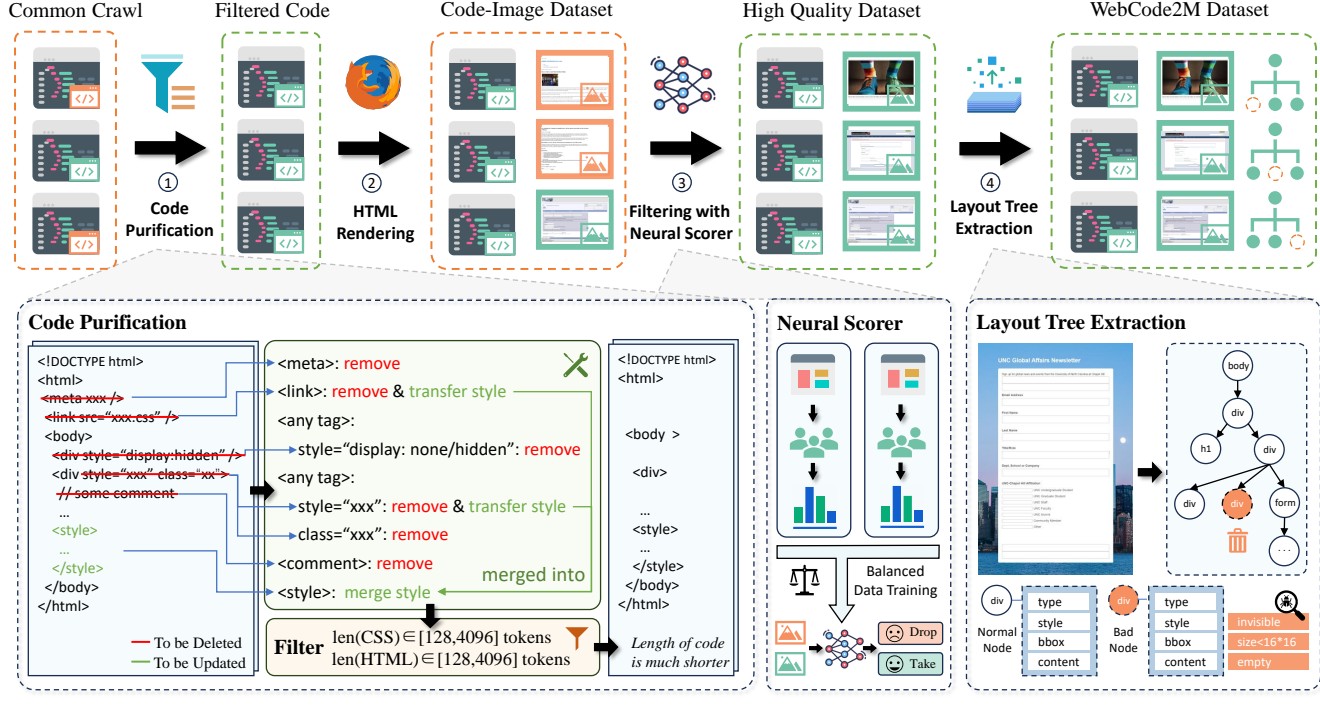

**Figure 1: The pipeline of constructing the WebCode2M dataset.**

code and establish a benchmark for fair comparison. Compared to the two fine-tuned baselines, Design2Code-18B [53] and WebSight VLM-8B [28], our model, fine-tuned from the smaller Pix2Struct-1.3B, outperforms both across all evaluation metrics, including CLIP-based visual similarity [44], low-level appearance accuracy [53], and our proposed TreeBLEU to measure the structural hierarchy recall. We also benchmark a broad array of general-purpose MLLMs, including the LlaVa family [34], CogAgent-Chat-18B [25], GPT-4V, GPT-4o [41], Gemini [13], and Claude [1]. Experimental results show that WebCoder outperforms these models across most evaluation metrics. The only exception is GPT-4o, which achieves higher similarity in CLIP and visual appearance but has a lower substructure recall rate.

**Contributions.** The primary contributions of our work are summarized as follows:

- **New Dataset.** To the best of our knowledge, WebCode2M is the first real-world and large-scale dataset tailored to empower MLLMs in the domain of generating webpage code from high-fidelity images.
- **Comprehensive Benchmark.** We fine-tune an MLLM, named WebCoder, on our WebCode2M dataset and evaluate it through a comprehensive set of experiments alongside other fine-tuned baselines. Experimental results demonstrate the effectiveness of the dataset in enabling MLLMs to automatically generate code from webpage designs. Additionally, we introduce a novel metric, TreeBLEU, to measure the structural hierarchy recall.
- **Open-Source Resources.** We open-source the code base, the dataset, and the new benchmark model, making them freely available to the research and developer communities, for further

innovation in automating front-end engineering. The resources are available at https://webcode2m-anonymous.github.io.

## 2 WebCode2M: The Dataset

This section details the construction process of the WebCode2M dataset, outlines its ethical compliance, describes the dataset partitioning, and highlights its key characteristics.

### 2.1 Dataset Construction

The aim of this study is to curate a dataset that facilitates training neural models to generate code from webpage designs. As large-scale human-designed screenshots are hard to collect manually, we opt to reversely generate screenshot image from a curated open-source web dataset via rendering the webpage code. Figure 1 illustrates the pipeline for constructing WebCode2M, encompassing steps such as code purification, HTML rendering, filtering with a neural scorer, and layout tree extraction.

**Raw Data Collection.** We build our dataset on top of the Common Crawl dataset [2], a comprehensive collection of global webpage data spanning from 2013 to the present, updated monthly through web crawling. Previously, the Common Crawl dataset is primarily used for pre-training models on text-based tasks. Due to our computational resource and download speed constraints, we randomly sample approximately 0.5 billion webpages from the first segment of the CC-MAIN-2023-50 version, which contains about 3.35 billion webpages, as our initial data. We then download the external CSS code for each HTML file and integrate it into the HTML text.

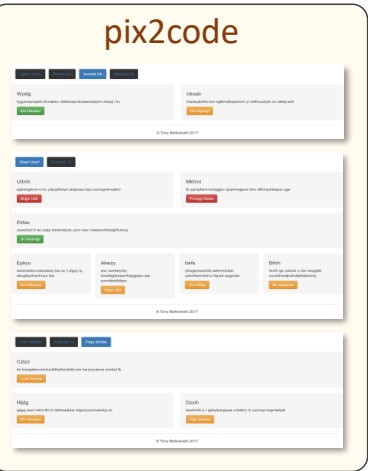 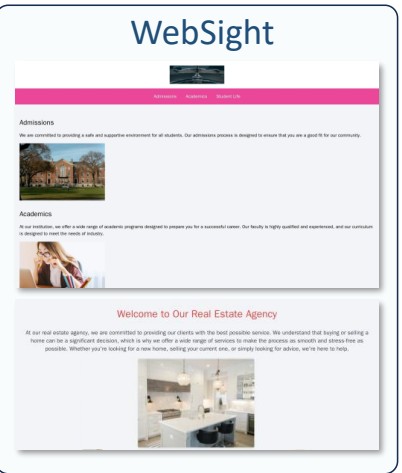 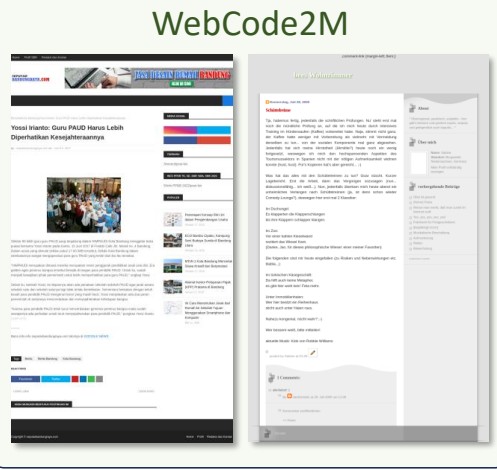

**Figure 2: Representative screenshots of webpages in WebCode2M and other datasets. From left to right are Pix2code, WebSight, and our WebCode2M dataset. Compared to the first two artificially synthesized datasets, ours is derived from real-world online websites, showcasing significantly greater diversity in elements, content, colors, and structural layouts.**

**Code Purification.** Our investigation reveals that webpage code from online websites is often lengthy and includes redundant elements, which significantly impairs the model's ability to accurately learn the correlation between webpage code and screenshots [28]. To ensure data quality, we meticulously clean the combined HTML and CSS text adhering to the following steps.

- Quick length filtering. Furthermore, we filter out samples that are either excessively long or too short. This is because parsing errors or other issues often lead to excessively short HTML or CSS code, while excessively long input contexts significantly slow down our training and inference procedures. Specifically, we employ a rapid filtering method based on code length, measured by the number of characters. Assuming that one word is approximately five characters long, we establish length ranges for HTML and CSS code between $[128 \times 5, 2048 \times 5]$ characters and $[128 \times 5, 4096 \times 5]$ characters, respectively. Webpages that fall outside these ranges are filtered out.

- Redundant code elements cleansing. The code samples in the raw dataset may include redundant elements, such as comments and hidden elements, as well as components that do not directly affect the rendering of static HTML pages. To address this, we propose removing the following contents from both HTML and CSS code: comments, `<meta>` and `<script>` tags, hidden elements (hidden, zero-sized, or outside the display range), attributes not in (`class`, `id`, `width`, `height`, `style`, `src`) of all HTML elements, and CSS styles that are not effective in the HTML code.

**HTML Rendering for Screenshot Generation.** After cleansing the data, we generate webpage screenshots from the combined HTML and CSS code. This process is implemented using Playwright [4], a headless browser automation tool that allows us to render webpages and capture high-fidelity screenshots. By simulating a real browser environment, Playwright ensures that the rendered webpage accurately reflects the appearance of the HTML and CSS code. This process is highly time-consuming, accounting

for roughly 80% of the total processing time, which spans approximately one month.

**Filtering with a Neural Scorer.** In our empirical data analysis, we observe that a considerable proportion of the generated screenshots exhibit deficiencies in aesthetics, as shown in Figure 9 (See Appendix). These low-quality screenshots are generally attributed to incompletely loaded pages resulted from various factors, for instance, invalid image links, and cases where the content is mainly composed of textual content. The presence of flawed screenshots can compromise the overall quality of the dataset, necessitating a rigorous filtering of the acquired data. Given the large volume of our dataset, manually screening all the data is impractical. Therefore, we train a classification model to serve as a neural scorer, assessing the screenshots and subsequently eliminating samples that fall below a specified score threshold.

In practice, we devise an annotating tool (See Figure 10 in Appendix) and manually annotate a subset of the generated screenshots. The scoring criteria are thoughtfully crafted, and each criterion satisfied will be awarded one point: (1) Normal webpage layout (human-designed layout, not simple auto single-column arrangement); (2) Normal webpage styling (elements like lists and blocks are styled, not using default styles); (3) No excessive blank areas; (4) Rich color combinations; and (5) Good aesthetic appearance. During the manual annotation process, we invite six annotators who hold a Bachelor's degree in Computer Science and have at least three years of web development experience. We then divide them into two groups to perform consensus annotation, where annotators within each group evaluate the same data. This annotation strategy minimizes the influence of subjective factors on the scoring results. The annotation process takes approximately two weeks for all the participants, ultimately yielding 10,000 manually scored data entries. The detailed annotation procedure is presented in Appendix C.

The score distribution of the manually labeled subset is depicted in Figure 4 (inner circle). The statistics reveal that 80% of the data

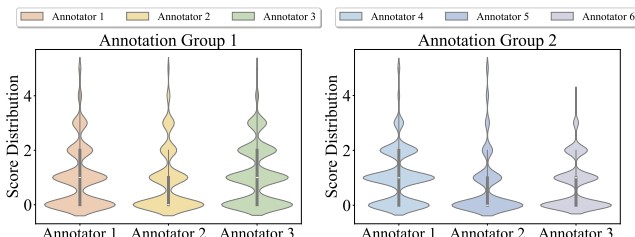

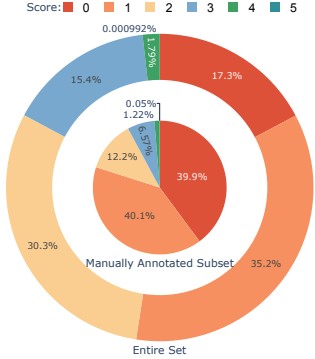

**Figure 3: Score distributions of annotators in two groups.**

**Figure 4: Score distribution of the manually annotated subset (inner ring) and the entire dataset (outer ring) before score-based filtering.**

fell within the low-quality category, scoring between 0 and 1. Conversely, only 20% of the entries, exhibiting scores of 2 or higher, demonstrate a commendable level of structural integrity and aesthetic appeal. We also conduct a consistency analysis of all the data from the annotators (See Figure 3). Although certain differences exist among annotators within the same group, the overall trend remains similar. By averaging scores within the same group, the impact of subjectivity is significantly reduced. Utilizing the rated data, we train a *ResNet-50* [23] model to serve as a scorer, predicting the score of input screenshots. This scorer achieves 75% accuracy on the test portion of the manually scored subset and nearly 90% accuracy in binary classification, determining whether the score is greater than or equal to two. Using this scorer, we remove samples with scores less than two, which accounted for 52.5% of the entire raw dataset (as shown in the outer circle of Figure 4).

**Layout Tree Extraction.** Considering that the webpage's layout defines the spatial arrangement and relationships between UI components, it can serve as a critical source of information. If available, the layout can act as a training target for the model, facilitating code generation by guiding the model to understand not only the structure of the webpage but also the precise positioning of elements. Thus, each data instance in our dataset is upgraded to a triplet: (webpage code, design image, layout). The layout, represented by the bounding boxes (BBox) of HTML elements, includes key information such as the size, location, and hierarchy of page components. This additional layout data will aid the model in learning to generate the webpage DOM tree structure more accurately.

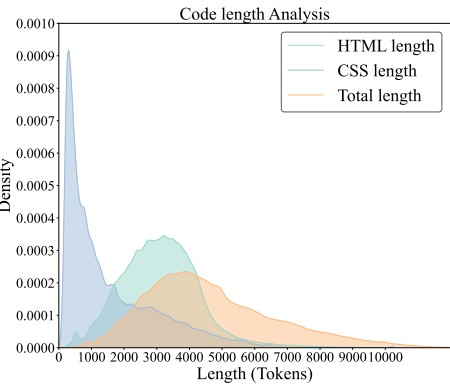

**Figure 5: Length density of the WebCode2M dataset.**

## 2.2 Ethical Compliance

Since our dataset is sourced from online webpages, it may contain content that is inappropriate for public release, such as explicit material or violent content. To mitigate ethical concerns regarding potential negative impacts, such as the misuse of models trained on this dataset, we perform additional filtering steps. Specifically, we apply an image filter to the screenshots and a profanity filter to the web text. Only samples that passed both filters are retained. Detailed filtering procedures are provided in Appendix F.

**Table 1: Dataset Partition.**

| Subset | Purpose | Size | Length (Tokens) |
|---|---|---|---|
| WebCode2M | Training | 2,563,905 | [368,16668] |
| WebCode2M-Short | Testing | 256 | [551, 2045] |
| WebCode2M-Mid | Testing | 256 | [2052, 4085] |
| WebCode2M-Long | Testing | 256 | [4098, 10990] |

## 2.3 Dataset De-Duplication and Partition

To support the use of our dataset in both fine-tuning models for webpage code generation and evaluating their performance, we organize the data into well-structured partitions. After using the hash codes of the screenshots to quickly de-duplicate the refined dataset, which comprises millions of entries, we sample approximately two thousand entries with a score above 4 as our candidate test dataset. The remaining 2.56 million entries serve as the training dataset. For the candidate test dataset, we further remove duplicates using CLIP [44] similarity and conduct a manual inspection on each entry.

Furthermore, we partition the test subsets based on code length to assess the model's code generation capability across varying levels of difficulty. As illustrated in Figure 5, the dataset shows a wide range of data length variations. Specifically, we use two length thresholds (i.e., 2048 and 4096), to select 256 entries from within three length ranges, thus creating three test subsets. We refer to them as WebCode2M-Short, WebCode2M-Mid, and WebCode2M-Long. Table 1 summarizes the overall statistics and provides detailed length statistics of WebCode2M for both training and testing.

## 2.4 Dataset Characteristics

Upon acquiring the final dataset WebCode2M, we conduct an analysis to identify several key characteristics. To quantitatively assess

**Table 2: A statistical comparison between our dataset and all the publicly available datasets. The statistical data of WebSight and Design2Code is referred to [53].**

| Dataset | Pix2code | WebSight | Design2Code | WebCode2M | WebCode2M_Short | WebCode2M_Mid | WebCode2M_Long |
|---|---|---|---|---|---|---|---|
| Purpose | Training&Testing | Training | Testing | Training | Testing | Testing | Testing |
| Source | Synthetic | Synthetic | Real-World | Real-World | Real-World | Real-World | Real-World |
| Size (#samples) | 1742 | 0.8M | 484 | 2.5M | 256 | 256 | 256 |
| Avg. Len (#tokens) | 1316±177 | 647±216 | 31216±23902 | 5366±2393 | 2025±514 | 3750±765 | 7940±1853 |
| Avg. Tags | 52±8 | 19±8 | 158±100 | 184±77 | 81±34 | 144±61 | 222±81 |
| Avg. DOM Depth | 8±0 | 5±1 | 13±5 | 15±5 | 10±4 | 13±7 | 16±4 |
| Avg. Unique Tags | 17±0 | 10±3 | 22±6 | 24±6 | 18±4 | 21±5 | 26±5 |

the diversity and quality of our dataset, we employ the same statistical metrics used in Design2Code, facilitating a comparison with other datasets. The results are presented in Table 2. Specifically, Avg. Len represents the token length as determined by the GPT-2 tokenizer [3]; Avg. Tags indicates the total number of tags in the HTML code; Avg. Unique Tags denotes the count of distinct tags in the HTML code; and Avg. DOM Depth signifies the maximum depth of the HTML's DOM Tree.

**Superior Diversity.** From Table 2, it is apparent to see that our dataset contains a significantly greater number and diversity of HTML tags and exhibits a more intricate DOM tree structure compared to the pix2code and WebSight datasets. This suggests that our dataset, sourced from real-world webpages, offers a remarkable diversity advantage over synthetic datasets generated by LLMs like WebSight. Design2Code, which also utilizes real-world data through the C4 dataset [45] from the Common Crawl corpus, exhibits a comparable distribution across these metrics, underscoring the benefits of real-world data in capturing the complexity of actual webpages. Moreover, this comparison highlights significant deviations in data attribute distributions between LLM-generated datasets and real webpages.

Figure 2 presents several representative screenshots from the datasets (excluding Design2Code). The pix2code dataset comprises basic block elements and text-based UI elements, suitable for both Android and iOS UIs. In contrast, WebSight consists of structurally simple webpages. Our dataset, on the other hand, closely mirrors typical real webpages, featuring a variety of layout structures and rich elements such as images. Additionally, our dataset captures webpages in a diverse range of languages (See Figure 12 in the Appendix).

**Large Scale and High Quality.** Compared to pix2code and Design2Code, which contain only a few thousand or fewer data samples, and WebSight, which includes 0.8 million samples, our dataset is significantly larger, comprising 2.56 million samples. This dataset includes both a comprehensive training dataset and a high-quality test dataset, making it much larger in scale. Notably, compared to Design2Code, our WebCode2M significantly reduces the average code length to about one-tenth of its original size, while maintaining the diversity and quantity of HTML tags, thereby preserving the high quality of the dataset.

## 3 Benchmarking

We introduce a baseline model based on the ViT, named WebCoder, and establish a benchmark for fair comparison.

### 3.1 WebCoder: A Reference Baseline

To demonstrate the potential of our dataset in enhancing automatic webpage code generation, we fine-tune a ViT model to establish a new baseline for translating design images into HTML/CSS code. Specifically, we select Google's Pix2Struct-1.3B [29] as our base model. This model, based on the ViT architecture, has been pre-trained on webpage code derived from URLs in the C4 dataset [45]. Pix2Struct-1.3B is notable for its robustness to extreme aspect ratios and ability to adapt dynamically to changes in sequence length and resolution. Furthermore, it rescales images by distorting the aspect ratio to preserve original image information, facilitating the processing of variable resolutions. We conduct a full fine-tuning of the pre-trained Pix2Struct on our training dataset, resulting in our model, WebCoder.

### 3.2 Setup and Baselines

Our evaluation experiments focus on two primary Research Questions (RQs):

**RQ1: The effectiveness of the training dataset.** To investigate the ability of our training dataset to empower MLLMs in webpage generation, we compare WebCoder with several state-of-art models which are also fine-tuned specifically for the webpage generation task:

- **WebSight VLM-8B** [28]. Hugginface's WebSight utilized its training dataset and the DoRA [35] mechanism to fine-tune a base VLM, which has been pre-trained on image/text pairs.
- **Design2Code-18B** [53]. Stanford's Design2Code is also fine-tuned on the WebSight dataset. It adopts CogAgent [25] as its base model and utilizes LoRA [26] as the finetuning method to accelerate the training process.
- **WebCoder\*.** Another Pix2Struct model in the same setting but trained on the WebSight dataset for comparative experiments.

**RQ2: Benchmarking on the test datasets.** We also introduce a broad array of the latest and most powerful general-purpose pre-trained MLLMs for benchmarking:

- **LLaVA Family** [34]. The LLaVA family consists of various MLLMs that connect a vision encoder and an LLM for general-purpose visual and language understanding. In our work, we introduce **LLaVA-v1.5-7B**, **LLaVA-onevision-0.5B**, and **LLaVA-onevision-7B** as the baselines. The prompt used for these models follows [8] and is detailed in Appendix A.
- **CogAgent-Chat-18B** [25]. CogAgent-Chat-18B is a general MLLM that supports both low- and high-resolution images and performs quite well on webpage navigation. We also input the

screenshot and a simple prompt *Write an HTML code* to generate the webpage as Design2Code does.

- **Commercial Models.** Some general commercial models have demonstrated impressive performance across various fields, proficient in both code generation and web understanding. Therefore, we introduce OpenAI's **GPT-4V** and **GPT-4o** [41], Google DeepMind's **Gemini** [13], and Anthropic's **Claude** [1] as baselines. The prompt for these models also follows [8], detailed in Appendix A.

Although previous work [53] suggests that multi-round generation methods (*e.g.*, self-correction) may outperform one-pass generation, baseline models such as Design2Code-18B and WebSight VLM-8B are fine-tuned with one-pass dataset and only support one-pass generation. Therefore, to ensure a fair comparison, all baselines will employ the one-pass generation strategy.

## 3.3 Evaluation Metrics

This section presents the evaluation metrics used in our work, including measurements for visual similarity and structural similarity, as well as several classic metrics.

**Visual Similarity Measurement.** We adopt CLIP [44] similarity and *Visual Score* [53] as two major metrics to assess visual similarities between the generated webpage page and ground truth. CLIP similarity is derived from calculating the cosine value of two images' latent vectors encoded by CLIP, which measures the overall visual similarity of two images. *Visual Score* is utilized to measure the matching degree of low-level elements in terms of appearance, calculating the average scores of the matching ratio between the reference and candidate blocks, as well as the similarity at four block levels in terms of color, text, CLIP, and position.

**Structure Similarity Measurement.** Skeletons of webpage code that determines the layout and appearance of the page, also known as the HTML DOM Tree, can also serve as a metric to evaluate structural similarity that compares ground truth (for instance, during the training phase or when the target code is provided in the inference stage) and the DOM Tree of the generated code. Inspired by [47], we propose a new metric **TreeBLEU** to evaluate the matching degree of the generated HTMLs' DOM tree (without terminal nodes that contain tags' attributes, *e.g.*, content and style) compared to the ground truth.

TreeBLEU is defined as the proportion of all 1-height subtrees (See Algorithm 1) in a given tree that can be matched with that of a reference tree. Let $S(.)$ be the set of 1-height subtrees, then it can be formulated as:

$$\text{TreeBLEU} = \frac{|S(t) \cap S(t_r)|}{|S(t_r)|},$$

where $t$ and $t_r$ denote the given tree and the reference tree, respectively. Different from htmlBLEU [54], a hybrid metric composed of four scores (the detailed definition of which is not available), our TreeBLEU focuses on the similarity of HTML DOM Tree in an integrated manner, as detailed in Appendix B.

**Classic Metrics.** We also assess the experimental results with several traditional metrics. Although these indicators primarily originate from Natural Language Processing (NLP) and some Computer Vision (CV) tasks, making them potentially less applicable to

---

**Algorithm 1** Get All 1-height Subtrees of DOM tree.

---

**Require:** A multiway tree node $root(childs, name)$
**Ensure:** A set $S$ of all 1-height subtrees
1: Initialize an empty set $S$
2: **function** TRAVERSE($node, S$)
3:    **if** number of children of $node \neq 0$ **then**
4:       Initialize an empty string $subtree$
5:       Append $node.name$ to $subtree$
6:       **for** each child $c$ in $node.childs$ **do**
7:          Append $c.name$ to $subtree$
8:       **end for**
9:       Add $subtree$ to $S$
10:    **end if**
11:    **for** each child $c$ in $node.childs$ **do**
12:       TRAVERSE($c, S$)
13:    **end for**
14: **end function**
15: TRAVERSE($root, S$)

---

our context, they still provide valuable insights into specific performance aspects. The conventional metrics we utilize include widely accepted evaluation indicators such as BLEU, ROUGE-I, MSE, and SSIM [59]. Detailed results can be found in Table 8 (See Appendix).

## 3.4 Implementation Details

We configure the model to process a maximum of 1,024 patches, representing the upper limit of image segments it can handle and set the maximum sequence length to 2,048 tokens. These settings are selected to balance training and inference speed with task accuracy. Due to GPU memory limitations, we set the batch size as 1 during training. In the initial phase of training, we fine-tune the model on a subset of our WebCode2M dataset, with a sequence length capped at 2,048 tokens. This phase consisted of three training epochs, totaling 90,000 iterations, with a maximum learning rate of 5e-5 and a cosine learning rate scheduler. The primary objective was to equip the model with the ability to generate code from visual inputs. Subsequently, we refine our approach by decreasing the maximum learning rate to 1e-5 and performing an additional three epochs of fine-tuning on a subset of the dataset, featuring a reduced sequence length of 1,024 tokens and consisting of 10,000 iterations. All the experiments are run on a Linux server equipped with 4 NVIDIA A100 80G GPUs.

## 3.5 Effectiveness of the Training Dataset (RQ1)

Table 3 presents the performance of WebCoder both on the WebSight and WebCode2M datasets, compared to other benchmark models on the WebSight dataset. From this figure, we can observe that our method consistently outperforms all specialized baselines across all three metrics on the real-world test dataset, noting that these specialized models were fine-tuned on the WebSight dataset. Comparative experiments also demonstrate that the base model, Pix2Struct, achieves a significant performance boost when fine-tuned on our training dataset compared to WebSight. For TreeBLEU—a metric measuring the recall of 1-height subtrees in the target DOM tree—our approach surpasses both specialized and

**Table 3: The performance comparison among the specialized models (the best is marked in bold).**

| Model | Training Dataset | WebCode2M-Short | | | WebCode2M-Mid | | | WebCode2M-Long | | |
|---|---|---|---|---|---|---|---|---|---|---|
| | | Visual | CLIP | TreeBLEU | Visual | CLIP | TreeBLEU | Visual | CLIP | TreeBLEU |
| WebSight VLM-7B | WebSight | $.57_{\pm.24}$ | $.69_{\pm.12}$ | $.03_{\pm.04}$ | $.52_{\pm.23}$ | $.67_{\pm.11}$ | $.03_{\pm.04}$ | $.48_{\pm.27}$ | $.64_{\pm.11}$ | $.03_{\pm.03}$ |
| Design2Code-18B | WebSight | $.75_{\pm.14}$ | $.68_{\pm.10}$ | $.04_{\pm.05}$ | $.69_{\pm.23}$ | $.70_{\pm.10}$ | $.05_{\pm.05}$ | $.61_{\pm.28}$ | $.68_{\pm.10}$ | $.06_{\pm.03}$ |
| WebCoder *-1.3B | WebSight | $.42_{\pm.32}$ | $.68_{\pm.11}$ | $.06_{\pm.06}$ | $.36_{\pm.30}$ | $.67_{\pm.11}$ | $.04_{\pm.04}$ | $.38_{\pm.29}$ | $.65_{\pm.11}$ | $.04_{\pm.04}$ |
| WebCoder-1.3B | WebCode2M | $\mathbf{.78_{\pm.25}}$ | $\mathbf{.73_{\pm.13}}$ | $\mathbf{.35_{\pm.17}}$ | $\mathbf{.69_{\pm.19}}$ | $\mathbf{.71_{\pm.10}}$ | $\mathbf{.22_{\pm.11}}$ | $\mathbf{.65_{\pm.21}}$ | $\mathbf{.69_{\pm.12}}$ | $\mathbf{.15_{\pm.07}}$ |

**Table 4: Benchmarking performance of several general-purpose MLLMs using the WebCode2M. (the best is marked in bold).**

| Model | WebCode2M-Short | | | WebCode2M-Mid | | | WebCode2M-Long | | |
|---|---|---|---|---|---|---|---|---|---|
| | Visual | CLIP | TreeBLEU | Visual | CLIP | TreeBLEU | Visual | CLIP | TreeBLEU |
| LLaVA-v1.5-7B | $.43_{\pm.27}$ | $.60_{\pm.33}$ | $.07_{\pm.05}$ | $.21_{\pm.28}$ | $.29_{\pm.38}$ | $.05_{\pm.04}$ | $.19_{\pm.27}$ | $.28_{\pm.37}$ | $.04_{\pm.03}$ |
| LLaVA-onevision-0.5B | $.24_{\pm.31}$ | $.62_{\pm.11}$ | $.06_{\pm.03}$ | $.28_{\pm.31}$ | $.61_{\pm.10}$ | $.05_{\pm.03}$ | $.22_{\pm.29}$ | $.59_{\pm.11}$ | $.03_{\pm.02}$ |
| LLaVA-onevision-7B | $.34_{\pm.32}$ | $.63_{\pm.10}$ | $.08_{\pm.07}$ | $.30_{\pm.30}$ | $.64_{\pm.09}$ | $.06_{\pm.06}$ | $.30_{\pm.30}$ | $.61_{\pm.10}$ | $.04_{\pm.04}$ |
| CogAgent-Chat-18B | $.46_{\pm.31}$ | $.68_{\pm.11}$ | $.01_{\pm.03}$ | $.40_{\pm.31}$ | $.66_{\pm.10}$ | $.01_{\pm.02}$ | $.39_{\pm.30}$ | $.65_{\pm.10}$ | $.01_{\pm.01}$ |
| Gemini | $.35_{\pm.41}$ | $.75_{\pm.10}$ | $\mathbf{.16_{\pm.10}}$ | $.38_{\pm.40}$ | $.74_{\pm.11}$ | $\mathbf{.15_{\pm.08}}$ | $.34_{\pm.41}$ | $.73_{\pm.10}$ | $\mathbf{.14_{\pm.06}}$ |
| Claude | $.52_{\pm.43}$ | $.77_{\pm.10}$ | $.13_{\pm.08}$ | $.35_{\pm.42}$ | $.76_{\pm.09}$ | $.14_{\pm.08}$ | $.37_{\pm.43}$ | $.74_{\pm.09}$ | $.13_{\pm.06}$ |
| GPT-4V | $.68_{\pm.32}$ | $.74_{\pm.10}$ | $.12_{\pm.07}$ | $.65_{\pm.33}$ | $.71_{\pm.10}$ | $.11_{\pm.06}$ | $.62_{\pm.35}$ | $.67_{\pm.10}$ | $.10_{\pm.05}$ |
| GPT-4o | $\mathbf{.85_{\pm.16}}$ | $\mathbf{.77_{\pm.10}}$ | $.15_{\pm.09}$ | $\mathbf{.81_{\pm.20}}$ | $\mathbf{.77_{\pm.09}}$ | $.13_{\pm.08}$ | $\mathbf{.82_{\pm.18}}$ | $\mathbf{.74_{\pm.09}}$ | $.11_{\pm.05}$ |

general-purpose models, indicating that our model better reflects real-world node types and substructures. Additionally, on the two visual similarity metrics—visual score and CLIP similarity—our model exceeds most general-purpose models and either matches or outperforms GPT-4V. Collectively, these results demonstrate that our dataset offers greater practical potential than synthetically generated datasets and suggest that our proposed training dataset can effectively unleash the potential of MLLMs in webpage generation.

## 3.6 Benchmarking on the Test Datasets (RQ2)

Table 4 benchmarks the performance of several general-purpose MLLMs using the WebCode2M test dataset. From this figure, we can observe several interesting findings: **(1) Generating lengthy code is challenging.** Almost all metrics for nearly all models drop significantly as the target code length increases. For example, as the dataset transitions from WebCode2M-short to WebCode2M-mid and finally to WebCode2M-long, the highest TreeBLEU score for specialized models drops from 0.35 to 0.15, the highest CLIP similarity decreases from 0.73 to 0.69, and the highest Visual Score declines from 0.78 to 0.65. **(2) Model size matters.** In LLaVA family, several models show a significant improvement across all metrics as model parameters increase, with LLaVA-v1.5-7B and LLaVA-onevision-7B achieving the best performance, while LLaVA-onevision-0.5B performs poorly across all metrics, indicating that MLLMs require more parameters to achieve better results in webpage generation tasks. **(3) Most general-purpose MLLMs struggle with webpage code generation.** Among these models, only GPT-4V matches the performance of our model trained on WebCode2M, while GPT-4o significantly outperforms all other models. All remaining general-purpose models generally underperform compared to specialized models, with consistently low scores across all metrics.

Notably, GPT-4o significantly outperforms all specialized and other general-purpose MLLMs across all metrics. Moreover, its performance remains highly stable as the complexity increases, without showing significant degradation. For instance, as the dataset

transitions from WebCode2M-short to WebCode2M-mid and finally to WebCode2M-long, its visual score, from 0.85 to 0.81 and then to 0.82. However, our goal is not to propose a dataset that allows small specialized models to surpass super MLLMs with hundreds of billions of parameters, as that would be unrealistic. Instead, **our aim is to assist MLLMs in the webpage generation task and enable smaller models to achieve competitive performance**.

## 4 Related Work

Generating code from webpage designs is essentially an image-to-code task, which primarily consists of two key components: image representation and code generation. We review the related works from the perspectives of image representation learning, code generation, and image-to-code.

**Image Representation Learning.** To obtain more suitable representations of images, early works proposed using Variational Autoencoders (VAEs) to generate latent vectors for images [27, 56]. Other researchers have explored employing contrastive learning to derive image encoders from large-scale training datasets [16]. The ViT was introduced to break down an image into sequences of fixed-size patches, applying a transformer to process them and accommodating variable resolutions [9]. In recent years, Diffusion Models (DM) [24] have achieved significant success in image representation learning, understanding, and generation tasks. To alleviate the computational burden of operating in pixel space, researchers proposed training DMs within the latent space of advanced pre-trained autoencoders [49]. Recently, SDXL[43] was developed, leveraging a three-times larger UNet [50] backbone and introducing a refinement model.

**Code Generation.** The development of code models has advanced significantly with the increasing availability of computational resources, evolving from small models with only a few million parameters to medium-sized models with billions of parameters, and ultimately to ultra-large models exceeding hundreds of billions. Early

works typically employed simple network architectures trained on small datasets for code generation tasks. For instance, some work [12, 20, 22] use RNNs [60] to treat code as token sequences, while others explored tree-structured neural networks [33, 37, 39] or graph neural networks [10, 11, 15] to capture the structural information of code. With the rise of transformer models, researchers began to explore supervised and unsupervised methods for training transformer-based models on large-scale codebases, such as GitHub. Notable works in this area include CodeBERT [18], CodeT5 [57], StarCoder [30], and AlphaCode [32], with AlphaCode even achieving an average ranking in the top 54.3% in simulated programming competitions on the Codeforces platform. More recently, the landscape of code generation has been significantly influenced by large language models (LLMs) such as CodeGen [38], CodeT5+[58], InCoder[19], GPT-3.5 [40], StarCoder [31], Code Llama [51], and WizardCoder [36].

**Image to Code.** Several early works have made pioneering contributions by focusing on generating code from simple images. For example, to reverse engineer program code from Graphical User Interfaces, [14] introduced pix2code, which is trained on a synthetic dataset of GUI screenshots and corresponding source code, including iOS, Android, or web-based GUI, to generate Domain-Specific Language code. Sketch2code [48] generates website code from wireframe sketches exploring two approaches: a computer vision-based method that detects elements and structures, and a deep learning-based method. With advances in computational power, some studies have explored using larger models to advance the image-to-code task. [61] addressed the challenge of *screen parsing* by predicting UI hierarchy graphs from screenshots, by using Faster-RCNN [46] to encode the screenshot images and employing an LSTM-based attention mechanism to construct graph nodes and edges. Pix2Struct [29], pre-trained by learning to generate simplified HTML from masked website screenshots, demonstrated substantial improvements in visual language understanding across nine tasks in four different domains. To address the challenges of rendering inefficiencies and non-differentiability in website generation, [55] employed reinforcement learning to fine-tune a vision-code transformer to minimize the visual differences between the original and generated HTML. While recent efforts have aimed at improving code generation from high-definition images, the results are still far from being ready for practical application.

Nowadays, several powerful commercial models have emerged, such as OpenAI's GPT-4V and GPT-4o [41], Google DeepMind's Gemini [13], and Anthropic's Claude [1]. These models have shown impressive performance across various tasks, including image understanding and code generation with the advantage of allowing continuous adjustment and optimization via chat. However, their performance in generating webpages from high-resolution images remains suboptimal.

## 5 Discussion

In this section, we discuss several practical challenges associated with our dataset when applied to webpage code generation and highlight several limitations that warrant further study.

### 5.1 Practical Challenges to Study

In the course of our research, we identify three practical challenges that need to be addressed to achieve the ideal generation of webpage code from design images. These challenges are presented here to guide future research: **(1) Lengthy code generation.** As shown in Table 2, despite our efforts to clean up noise in the webpage code, such as invisible elements, the HTML text remains lengthy, reflecting its complexity to some extent. This presents significant challenges to both the training effectiveness and efficiency of MLLMs. **(2) Capturing structural information of UI visions.** Given the potential overlap of sub-elements in images and the lack of distinct borders for some elements, extracting structured or hierarchical information from images presents a significant challenge. Our empirical study of GPT-4V reveals that, while it excels in capturing text and color from images when generating webpage code, it struggles with capturing the hierarchy of UI elements. Therefore, designing a model that is more proficient in generating the hierarchical structure for translating design diagrams into webpage code is a promising avenue. **(3) Generation of image elements.** All existing webpage code generation models fail to accurately reproduce image elements in the design visions, severely hindering their practical application. There is an urgent need for a framework capable of generating or extracting image elements from the original design and assembling them into the final webpage code.

### 5.2 Limitations and Future Directions

Firstly, although we employ a meticulously designed neural scorer to enhance data quality, this scoring method inherently contains a degree of subjectivity and achieves an accuracy of approximately 90%. Consequently, some low-quality data remain in the final dataset. However, we consider this acceptable given the trade-off between efficiency and quality, as manual screening of millions of data is impractical. Secondly, the results presented in Table 3 and Table 4 indicate that all models exhibit significant variance on certain metrics. This suggests that the model's generation capability is insufficiently stable and performs poorly on some test data, underscoring the need for a more robust framework to accomplish this task. Finally, because our dataset is sourced from crawled online data, it inevitably contains a minimal amount of inappropriate content, such as violent material, despite our extensive filtering efforts.

## 6 Conclusion

In this paper, we have proposed WebCode2M, the first real-world and large-scale dataset with layout information for generating webpage code from designs. This dataset consists of over 2.56 million samples for both training and testing. We have presented the detailed pipeline of dataset construction and conducted analysis on the curated dataset. The analysis results demonstrate the diversity of our dataset. We fine-tune an MLLM, named WebCoder, on our training dataset. Along with two visual measures and a structure metric, we evaluate our WebCoder with other baselines on the proposed dataset. The experiment results demonstrate that our dataset can better empower MLLMs to generate code from webpage designs. We believe that the dataset and benchmark proposed in this work can further advance research in this field.

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

# A   An Empirical Study

We conducted an empirical study by employing the state-of-the-art MLLM, GPT-4V, to generate webpage code from images in a one-pass manner. We refer to the prompt of a famous open-source project *screenshot-to-code* [8] on the GitHub, making only minor adjustments as shown in Figure 6.

We also investigated the capabilities of the pix2code and Pix2Struct models. The pix2code dataset was first divided into training and testing datasets with an 8:2 split. We then trained the pix2code model from scratch on the training dataset while simultaneously fine-tuning the Pix2Struct model. To quantitatively assess the performance of pix2code, Pix2Struct, and GPT-4V on the pix2code test dataset, we utilized two automated metrics.

Table 5 shows the comparison results of the performance of several baselines on the pix2code test dataset. From this table, it can be seen that ChatGPT-4V's performance in one-pass generation mode lags behind that of the pix2code model, and the model fine-tuned from Struct2Code, even when applied to the simplest pix2code dataset. This performance discrepancy is particularly evident in the generation of the HTML DOM tree structure, which is represented as TreeBLEU in Table 5.

To more vividly illustrate the shortcoming of GPT-4V in capturing the structural information in original screenshots, we choose a representative example (similar situations are often found in actual test results), as shown in Figure 7. It can be clearly observed from the figure that the webpage generated by GPT-4V does not reflect the hierarchical structure of the referenced image, but rather appears as a simple auto-sorted list. This situation is also frequently encountered during our experience with the *screenshot-to-code*. The primary explanation for this disparity may lay in the fact that both models have been fine-tuned on specific task datasets, while ChatGPT-4V has not.

**Table 5: The performance comparison on the pix2code test dataset. The pix2code model and Pix2Struct model are both fine-tuned on the pix2code training dataset. The GPT-4V is prompted to generate webpage code in one-pass mode.**

| Model | TreeBLEU |
|---|---|
| pix2code-Beam | 0.98 |
| pix2code-Gready | 0.99 |
| Pix2Struct-282M | 0.79 |
| Pix2Struct-1.3B | 0.92 |
| GPT-4V | 0.09 |

# B   The Implementation of TreeBLEU

The core implementation of TreeBLEU includes two parts: the generation of a minimalist HTML DOM Tree and the Subtree Collection Algorithm.

- **Minimalist HTML DOM Tree.** The original HTML DOM Tree can be easily obtained by some tools such as the bs4 [6] package in Python. After acquiring the HTML DOM Tree, the next step is to remove the terminal nodes, the text nodes and image nodes, etc. The resulting DOM Tree consists only of the type names of HTML tags.
- **Subtree Collection Algorithm,** as shown in Algorithm 1.

Given that the subtree collections of the generated HTML DOM Tree and the reference are represented as sets of strings, subtree matching can be straightforwardly performed through string comparison. The TreeBLEU value can then be calculated using Equation 3.3. An example of subtree matching is illustrated in Figure 8.

# C   Details of Manual Annotation

## C.1   Motivating Examples

After performing extensive data cleansing and formatting on our large-scale web-crawled dataset—including length filtering, removal of unnecessary tags and attributes, and style consolidation—we conducted a manual sampling review. This review revealed that many instances still exhibited structural and aesthetic flaws (Figure 9), as evidenced by the rendered screenshots, often due to missing styles or broken image links. To address these issues, we implemented a web screenshot quality scoring system, trained on manually labeled data, to further refine the dataset through an additional, meticulous screening process.

## C.2   Scoring Tool

To enhance the efficiency of manual data labeling, we developed an image scoring tool using Gradio. As illustrated in Figure 10, the tool features a screenshot display area on the left and a scoring interface on the right. The right side includes selectable options for evaluating image quality, along with buttons for navigating data entries, labeling the current entry, and saving all annotations.

## C.3   Scoring Procedure

To ensure consistency in our final labeled dataset and reduce subjectivity among annotators, we developed a transparent grading system with five hierarchical criteria, ranging from basic to advanced, integrated into our labeling interface. Annotators are instructed to select the relevant options for each data entry, with the total number of selected options determining the score. The grading scale spans from 0 to 5 points, making it straightforward for annotators to apply and ensuring uniform data evaluation. The scoring process involves six annotators divided into two groups. Each group assesses a designated subset, and individual scores are averaged and rounded to the nearest whole number to produce the final score for each entry.

## C.4   Consistency Validation

We conduct a statistical analysis of the score distribution within two annotated cohorts, as shown in Figure 3. The graphical representation reveals noticeable disparities among the annotations, despite the consistency of the dataset and a generally aligned distribution across the three annotators within each group. This observation highlights the subjective nature of screenshot scoring. Therefore, adopting an averaging approach within each group proves to be an effective strategy for reducing the impact of subjectivity on scoring outcomes.

# D   More Statistics of Dataset

We perform a comprehensive statistical analysis on the textual content of webpages, and the findings are illustrated in Figure 12.



**Prompt for using GPT-4V in webpage generation task**

You are an expert Tailwind developer. You take screenshots of a reference webpage from the user, and then build single-page apps using Tailwind, HTML and JS.
- Make sure the app looks exactly like the screenshot.
- Make sure the app has the same page layout like the screenshot, i.e., the gereated html elements should be at the same place with the corresponding part in the screenshot and the generated html containers should have the same hierachy structure as the screenshot.
- Pay close attention to background color, text color, font size, font family, padding, margin, border, etc. Match the colors and sizes exactly.
- Use the exact text from the screenshot.
- Do not add comments in the code such as "<!– Add other navigation links as needed –>" and "<!– ... other news items ... –>" in place of writing the full code. WRITE THE FULL CODE.
- Repeat elements as needed to match the screenshot. For example, if there are 15 items, the code should have 15 items. DO NOT LEAVE comments like "<!– Repeat for each news item –>" or bad things will happen.
- For images, use placeholder images from https://placehold.co and include a detailed description of the image in the alt text so that an image generation AI can generate the image later. In terms of libraries,
- Use this script to include Tailwind: <script src="https://cdn.tailwindcss.com"></script>
- You can use Google Fonts
- Font Awesome for icons: <link rel="stylesheet" href="https://cdnjs.cloudflare.com/ajax/libs/font-awesome/5.15.3/css/all.min.css"></link> Return only the full code in <html></html> tags.
- Do not include markdown """ or ""html" at the start or end.



**Figure 6: Prompt for using GPT-4V in webpage generation task.**

**Table 6: Licenses of the datasets in this work.**

| Dataset | License |
| --- | --- |
| pix2code | Apache 2.0 |
| Common Crawl | LIMITED license |
| WebSight | CC-BY-4.0 |
| Design2Code | MIT |
| WebCode2M | CC-BY-4.0 |

English constitutes approximately 50% of the corpus, with other languages such as Russian, German, Spanish, and French each representing nearly 5% of the total corpus. We also present the licenses of the datasets in our work in Table 6.

## E More Experimental Results

The Visual Score used in Table 3 and Table 4 is actually a composite metric, composed of five indicators: block, text, position, text color, and CLIP match. We list these detailed indicators in Table 7 for reference. Table 8 presents a detailed performance comparison based on traditional metrics.

## F Implementation of Filtering Inappropriate Content

The main steps and details of the cleansing process are as follows:
**Step 1: Filtering harmful images in screenshots.** We employ a widely used NSFW detector [7] from Hugging Face, which classifies images as either "normal" or "NSFW" (Not Safe for Work) with high accuracy. This detector is based on a Vision Transformer (ViT) model fine-tuned on a dataset containing both "safe" and "explicit" images. It predicts two scores: "normal" and "NSFW," with lower

NSFW scores indicating a lower probability of harmful content. A conservative threshold of 0.04 was adopted, meaning only samples with NSFW scores below this value were retained.

**Step 2: Multi-language harmful keyword filtering.** We apply harmful keyword filtering to the web text using two popular GitHub repositories: bad words 1 [52] and bad words 2 [5]. The first list contains dirty, naughty, obscene, and otherwise inappropriate words in multiple languages (*e.g.*, 'fuck', 'raping'), while the second serves as a supplementary list with additional sensitive or stop words. Samples with more than 20 occurrences of these bad words are removed. While normal web pages may occasionally contain a small number of inappropriate words, those with excessive amounts of inappropriate content, such as adult websites, tend to have significantly higher word counts.

The thresholds for both the NSFW score and the frequency of bad words were determined through experiments on the first data chunk 9. We selected thresholds that balanced maximizing sample retention with minimizing the presence of harmful content. To evaluate the effectiveness of the filters, we manually reviewed and documented the filtering results for the first chunk under different thresholds. The table below clearly indicates that an NSFW threshold of 0.04 produced the best results.

## G Page Samples Generated By WebCoder

In Figure 13, we present several sample pages generated by our WebCoder model to demonstrate its capabilities. The left side shows the original webpage screenshots from our WebCode2M dataset, while the right side displays the outputs generated by our model via end-to-end inference after being trained on this dataset. The

**Table 7: Experimental Breakdown on the Visual Scores.**

| Test Set | Model | Training | Block | Text | Position | Text Color | CLIP |
|---|---|---|---|---|---|---|---|
| WebCode2M-Short | CogAgent-Chat-18B | - | 0.25 (±0.33) | 0.54 (±0.45) | 0.41 (±0.35) | 0.45 (±0.42) | 0.67 (±0.29) |
| | WebSight VLM-8B | WebSight | 0.19 (±0.26) | 0.65 (±0.33) | 0.60 (±0.32) | 0.63 (±0.36) | 0.78 (±0.20) |
| | Design2Code-18B | WebSight | 0.65 (±0.31) | 0.92 (±0.16) | 0.72 (±0.16) | 0.68 (±0.24) | 0.80 (±0.10) |
| | LLaVA-v1.5-7B | - | 0.07 (±0.11) | 0.53 (±0.38) | 0.43 (±0.33) | 0.50 (±0.41) | 0.60 (±0.33) |
| | LLaVA-onevision-0.5B | - | 0.07 (±0.18) | 0.31 (±0.42) | 0.25 (±0.35) | 0.25 (±0.39) | 0.33 (±0.38) |
| | LLaVA-onevision-7B | - | 0.16 (±0.27) | 0.40 (±0.44) | 0.34 (±0.39) | 0.29 (±0.36) | 0.52 (±0.37) |
| | Gemini | - | 0.32 (±0.44) | 0.42 (±0.48) | 0.36 (±0.42) | 0.30 (±0.38) | 0.35 (±0.40) |
| | Claude | - | 0.51 (±0.47) | 0.57 (±0.48) | 0.52 (±0.44) | 0.48 (±0.43) | 0.50 (±0.41) |
| | GPT-4V | - | 0.64 (±0.39) | 0.79 (±0.37) | 0.67 (±0.32) | 0.59 (±0.34) | 0.67 (±0.31) |
| | GPT-4o | - | 0.84 (±0.28) | 0.95 (±0.17) | 0.85 (±0.16) | 0.79 (±0.20) | 0.81 (±0.15) |
| | WebCoder * | WebSight | 0.17 (±0.26) | 0.50 (±0.42) | 0.44 (±0.38) | 0.41 (±0.40) | 0.57 (±0.36) |
| | WebCoder | WebCode2M | 0.57 (±0.38) | 0.81 (±0.31) | 0.70 (±0.28) | 0.66 (±0.32) | 0.73 (±0.27) |
| WebCode2M-Mid | CogAgent-Chat-18B | - | 0.19 (±0.29) | 0.45 (±0.46) | 0.35 (±0.37) | 0.36 (±0.41) | 0.64 (±0.30) |
| | WebSight VLM-8B | WebSight | 0.13 (±0.20) | 0.61 (±0.32) | 0.56 (±0.32) | 0.59 (±0.35) | 0.73 (±0.25) |
| | Design2Code-18B | WebSight | 0.55 (±0.35) | 0.84 (±0.28) | 0.66 (±0.24) | 0.62 (±0.28) | 0.75 (±0.23) |
| | LLaVA-v1.5-7B | - | 0.03 (±0.07) | 0.26 (±0.37) | 0.22 (±0.33) | 0.23 (±0.36) | 0.29 (±0.38) |
| | LLaVA-onevision-0.5B | - | 0.06 (±0.16) | 0.35 (±0.43) | 0.28 (±0.36) | 0.29 (±0.40) | 0.41 (±0.39) |
| | LLaVA-onevision-7B | - | 0.11 (±0.22) | 0.34 (±0.42) | 0.27 (±0.34) | 0.28 (±0.36) | 0.51 (±0.38) |
| | Gemini | - | 0.39 (±0.45) | 0.46 (±0.48) | 0.38 (±0.41) | 0.28 (±0.35) | 0.39 (±0.41) |
| | Claude | - | 0.35 (±0.45) | 0.40 (±0.48) | 0.35 (±0.42) | 0.31 (±0.40) | 0.35 (±0.41) |
| | GPT-4V | - | 0.61 (±0.38) | 0.78 (±0.38) | 0.65 (±0.32) | 0.58 (±0.33) | 0.66 (±0.33) |
| | GPT-4o | - | 0.79 (±0.30) | 0.92 (±0.22) | 0.79 (±0.21) | 0.77 (±0.22) | 0.78 (±0.19) |
| | WebCoder * | WebSight | 0.10 (±0.18) | 0.45 (±0.42) | 0.40 (±0.38) | 0.33 (±0.35) | 0.51 (±0.38) |
| | WebCoder | WebCode2M | 0.46 (±0.34) | 0.83 (±0.22) | 0.72 (±0.22) | 0.70 (±0.26) | 0.77 (±0.21) |
| WebCode2M-Long | CogAgent-Chat-18B | - | 0.22 (±0.32) | 0.43 (±0.45) | 0.33 (±0.36) | 0.32 (±0.38) | 0.66 (±0.26) |
| | WebSight VLM-8B | WebSight | 0.13 (±0.22) | 0.59 (±0.37) | 0.50 (±0.33) | 0.54 (±0.36) | 0.65 (±0.30) |
| | Design2Code-18B | WebSight | 0.49 (±0.36) | 0.77 (±0.35) | 0.58 (±0.28) | 0.55 (±0.28) | 0.66 (±0.29) |
| | LLaVA-v1.5-7B | - | 0.04 (±0.11) | 0.23 (±0.36) | 0.19 (±0.30) | 0.20 (±0.32) | 0.28 (±0.37) |
| | LLaVA-onevision-0.5B | - | 0.06 (±0.16) | 0.27 (±0.40) | 0.22 (±0.34) | 0.22 (±0.35) | 0.35 (±0.37) |
| | LLaVA-onevision-7B | - | 0.11 (±0.22) | 0.35 (±0.42) | 0.28 (±0.35) | 0.28 (±0.37) | 0.47 (±0.37) |
| | Gemini | - | 0.35 (±0.45) | 0.40 (±0.48) | 0.34 (±0.40) | 0.28 (±0.34) | 0.33 (±0.40) |
| | Claude | - | 0.39 (±0.46) | 0.42 (±0.48) | 0.36 (±0.42) | 0.33 (±0.39) | 0.35 (±0.41) |
| | GPT-4V | - | 0.60 (±0.40) | 0.74 (±0.41) | 0.62 (±0.36) | 0.53 (±0.32) | 0.60 (±0.34) |
| | GPT-4o | - | 0.84 (±0.26) | 0.93 (±0.20) | 0.79 (±0.19) | 0.75 (±0.19) | 0.78 (±0.19) |
| | WebCoder * | WebSight | 0.14 (±0.23) | 0.47 (±0.42) | 0.40 (±0.36) | 0.32 (±0.32) | 0.55 (±0.35) |
| | WebCoder | WebCode2M | 0.46 (±0.34) | 0.80 (±0.25) | 0.66 (±0.23) | 0.62 (±0.26) | 0.73 (±0.22) |

**Table 8: The performance breakdown on classic metrics.**

| Model | WebCode2M-Short | | | | WebCode2M-Mid | | | | WebCode2M-Long | | | |
|---|---|---|---|---|---|---|---|---|---|---|---|---|
| | BLEU | Rough-1 | MSE | SSIM | BLEU | Rough-1 | MSE | SSIM | BLEU | Rough-1 | MSE | SSIM |
| CogAgent-Chat | $.33_{\pm.33}$ | $.40_{\pm.33}$ | $.35_{\pm.38}$ | $.59_{\pm.15}$ | $.27_{\pm.31}$ | $.31_{\pm.30}$ | $.32_{\pm.30}$ | $.58_{\pm.14}$ | $.22_{\pm.28}$ | $.26_{\pm.28}$ | $.36_{\pm.30}$ | $.60_{\pm.13}$ |
| WebSight VLM | $.17_{\pm.19}$ | $.20_{\pm.20}$ | $.32_{\pm.36}$ | $.62_{\pm.17}$ | $.10_{\pm.14}$ | $.11_{\pm.14}$ | $.37_{\pm.38}$ | $.59_{\pm.16}$ | $.11_{\pm.16}$ | $.13_{\pm.17}$ | $.39_{\pm.27}$ | $.61_{\pm.15}$ |
| Design2Code | $.50_{\pm.32}$ | $.56_{\pm.29}$ | $.39_{\pm.36}$ | $.58_{\pm.15}$ | $.45_{\pm.31}$ | $.51_{\pm.28}$ | $.35_{\pm.28}$ | $.56_{\pm.14}$ | $.33_{\pm.29}$ | $.41_{\pm.26}$ | $.37_{\pm.28}$ | $.61_{\pm.11}$ |
| LLaVA-v1.5-7B | $.04_{\pm.07}$ | $.06_{\pm.06}$ | $.35_{\pm.37}$ | $\mathbf{.65_{\pm.17}}$ | $.03_{\pm.05}$ | $.04_{\pm.05}$ | $.33_{\pm.35}$ | $\mathbf{.65_{\pm.14}}$ | $.03_{\pm.05}$ | $.05_{\pm.05}$ | $.34_{\pm.28}$ | $\mathbf{.66_{\pm.12}}$ |
| LLaVA-onevision-0.5B | $.11_{\pm.17}$ | $.17_{\pm.18}$ | $.38_{\pm.42}$ | $.53_{\pm.24}$ | $.07_{\pm.11}$ | $.12_{\pm.15}$ | $.35_{\pm.27}$ | $.49_{\pm.22}$ | $.08_{\pm.13}$ | $.13_{\pm.15}$ | $.37_{\pm.27}$ | $.55_{\pm.20}$ |
| LLaVA-onevision-7B | $.13_{\pm.21}$ | $.20_{\pm.22}$ | $.31_{\pm.33}$ | $.63_{\pm.15}$ | $.06_{\pm.14}$ | $.12_{\pm.16}$ | $.30_{\pm.28}$ | $.60_{\pm.14}$ | $.05_{\pm.12}$ | $.11_{\pm.15}$ | $.33_{\pm.27}$ | $.63_{\pm.12}$ |
| Gemini | $.64_{\pm.28}$ | $.75_{\pm.20}$ | $.32_{\pm.30}$ | $.62_{\pm.15}$ | $.68_{\pm.23}$ | $.74_{\pm.21}$ | $.33_{\pm.26}$ | $.60_{\pm.14}$ | $.63_{\pm.22}$ | $.69_{\pm.19}$ | $.39_{\pm.31}$ | $.61_{\pm.12}$ |
| Claude | $\mathbf{.74_{\pm.27}}$ | $\mathbf{.81_{\pm.22}}$ | $.29_{\pm.29}$ | $.61_{\pm.14}$ | $\mathbf{.74_{\pm.23}}$ | $\mathbf{.79_{\pm.20}}$ | $.30_{\pm.23}$ | $.59_{\pm.12}$ | $\mathbf{.68_{\pm.22}}$ | $\mathbf{.73_{\pm.18}}$ | $.33_{\pm.28}$ | $.61_{\pm.12}$ |
| GPT-4V | $.69_{\pm.28}$ | $.74_{\pm.24}$ | $.30_{\pm.25}$ | $.61_{\pm.14}$ | $.55_{\pm.32}$ | $.62_{\pm.28}$ | $.34_{\pm.29}$ | $.55_{\pm.12}$ | $.49_{\pm.29}$ | $.58_{\pm.23}$ | $.38_{\pm.28}$ | $.57_{\pm.11}$ |
| GPT-4o | $.73_{\pm.26}$ | $.80_{\pm.21}$ | $.27_{\pm.19}$ | $.60_{\pm.13}$ | $.70_{\pm.25}$ | $.74_{\pm.23}$ | $.27_{\pm.20}$ | $.58_{\pm.12}$ | $.67_{\pm.23}$ | $.71_{\pm.20}$ | $.30_{\pm.21}$ | $.60_{\pm.11}$ |
| WebCoder * | $.23_{\pm.23}$ | $.26_{\pm.22}$ | $.37_{\pm.33}$ | $.58_{\pm.17}$ | $.16_{\pm.17}$ | $.17_{\pm.18}$ | $.34_{\pm.28}$ | $.56_{\pm.16}$ | $.15_{\pm.17}$ | $.17_{\pm.17}$ | $.44_{\pm.33}$ | $.59_{\pm.15}$ |
| WebCoder | $.58_{\pm.26}$ | $.61_{\pm.24}$ | $\mathbf{.41_{\pm.44}}$ | $.59_{\pm.17}$ | $.46_{\pm.27}$ | $.47_{\pm.26}$ | $\mathbf{.39_{\pm.37}}$ | $.56_{\pm.16}$ | $.40_{\pm.26}$ | $.41_{\pm.24}$ | $\mathbf{.47_{\pm.44}}$ | $.58_{\pm.16}$ |

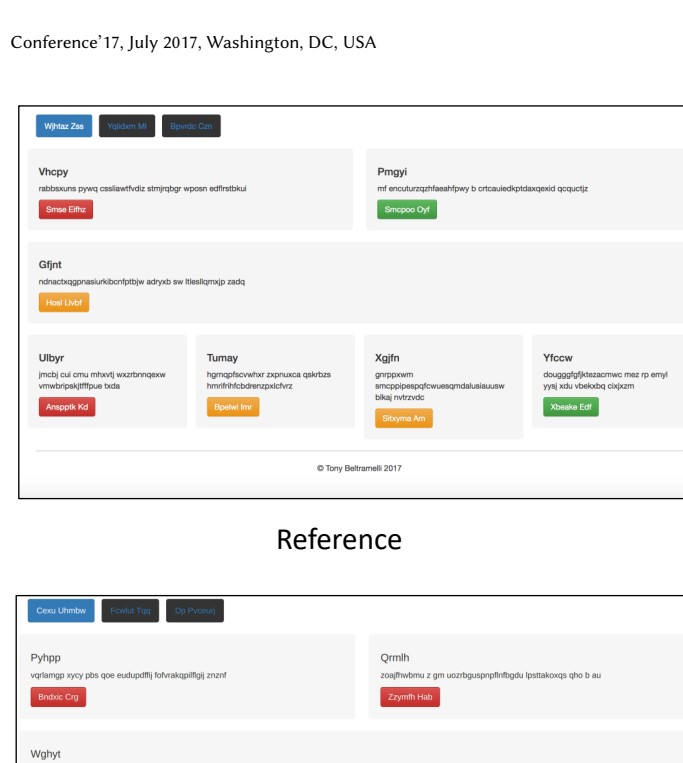

Reference

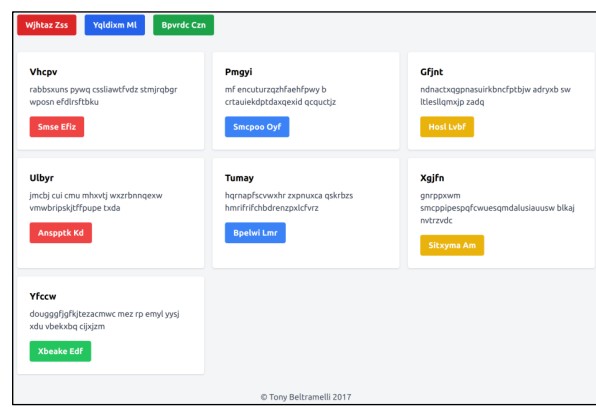

GPT-4V

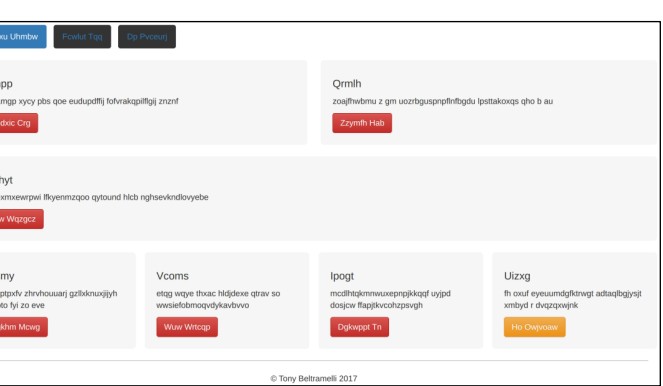

pix2code

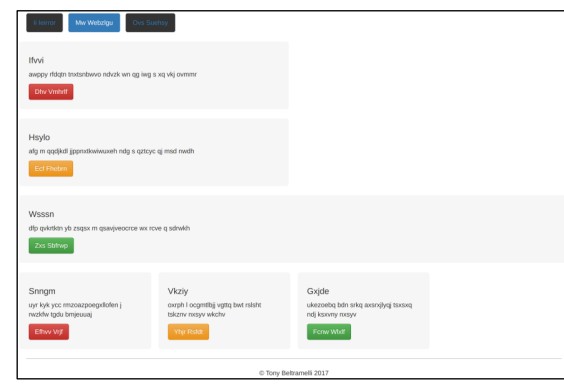

Pix2struct

Figure 7: A representative example on the pix2code test dataset.

Table 10: The filtering performance across different NSFW threshold values.

| Threshold | Total Num | Removed by nsfw filter | | | Retained after nfsw filter | | | | Retained after nfsw filter bad word filter | | | |
|---|---|---|---|---|---|---|---|---|---|---|---|---|
| | | total num | good num | miskill ratio | total num | bad num | retention ratio | toxic ratio | total num | bad num | retention ratio | toxic ratio |
| 0.02 | | 891 | 883 | 57.49% | 645 | 0 | 41.99% | 0.00% | 645 | 0 | 41.99% | 0.00% |
| 0.03 | | 324 | 318 | 20.70% | 1212 | 2 | 78.91% | 0.17% | 1210 | 0 | 78.78% | 0.00% |
| 0.04 | 1536 | 133 | 128 | 8.33% | 1403 | 3 | 91.34% | 0.21% | 1400 | 0 | 91.15% | 0.00% |
| 0.05 | | 70 | 66 | 4.30% | 1466 | 4 | 95.44% | 0.27% | 1463 | 1 | 95.25% | 0.07% |
| 1.00 | | 0 | 0 | 0.00% | 1536 | 8 | 100.00% | 0.52% | 1530 | 2 | 100.00% | 0.13% |

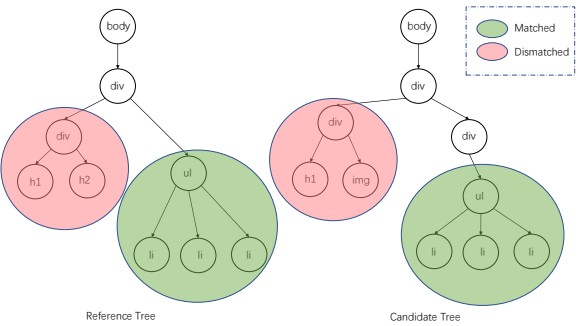

images were processed using OCR, extracted from the original screenshots, and embedded into the predicted HTML code. The results indicate that our model effectively preserves both structural and stylistic consistency while achieving high accuracy in text content prediction.

Figure 8: A subtree matching example in the metric of Tree-BLEU.

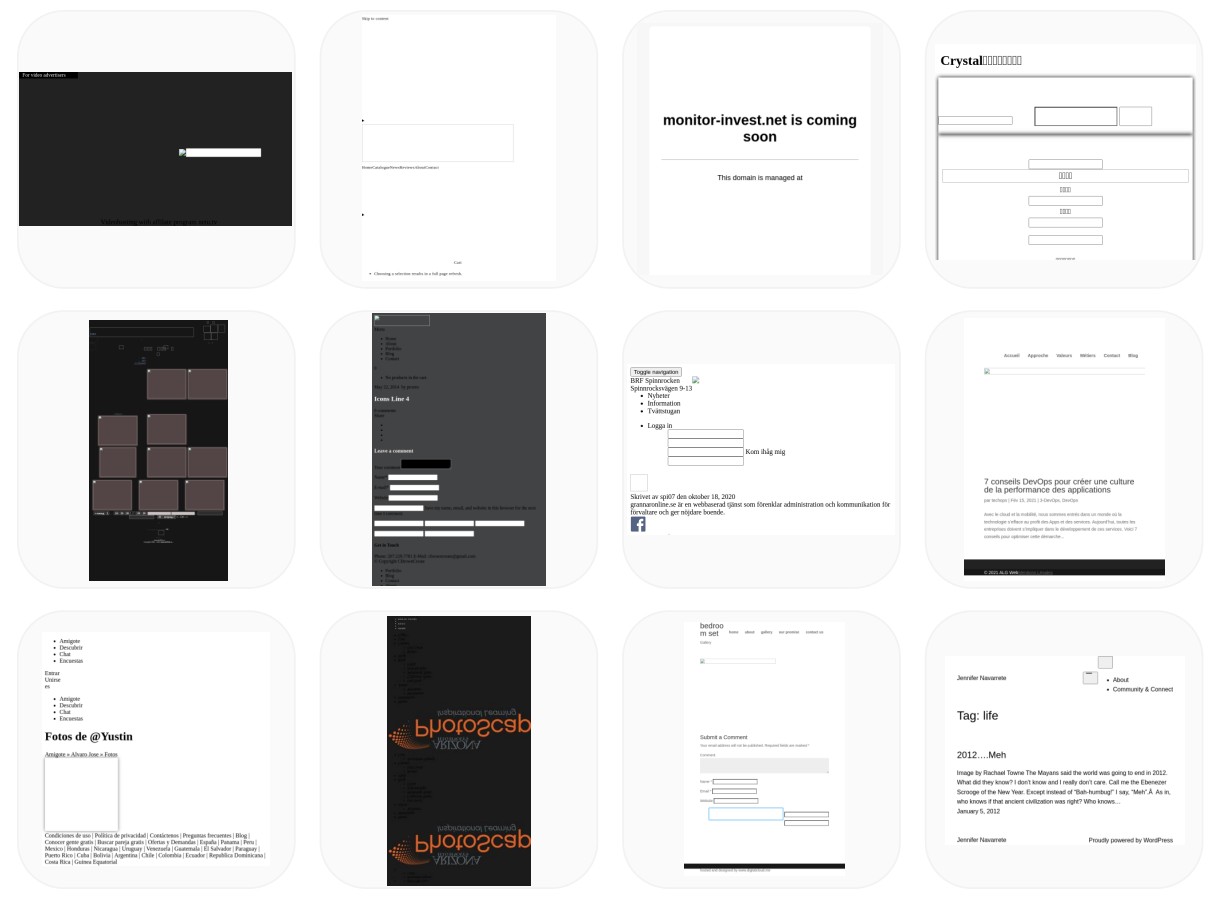

**Figure 9: Low-quality samples.**

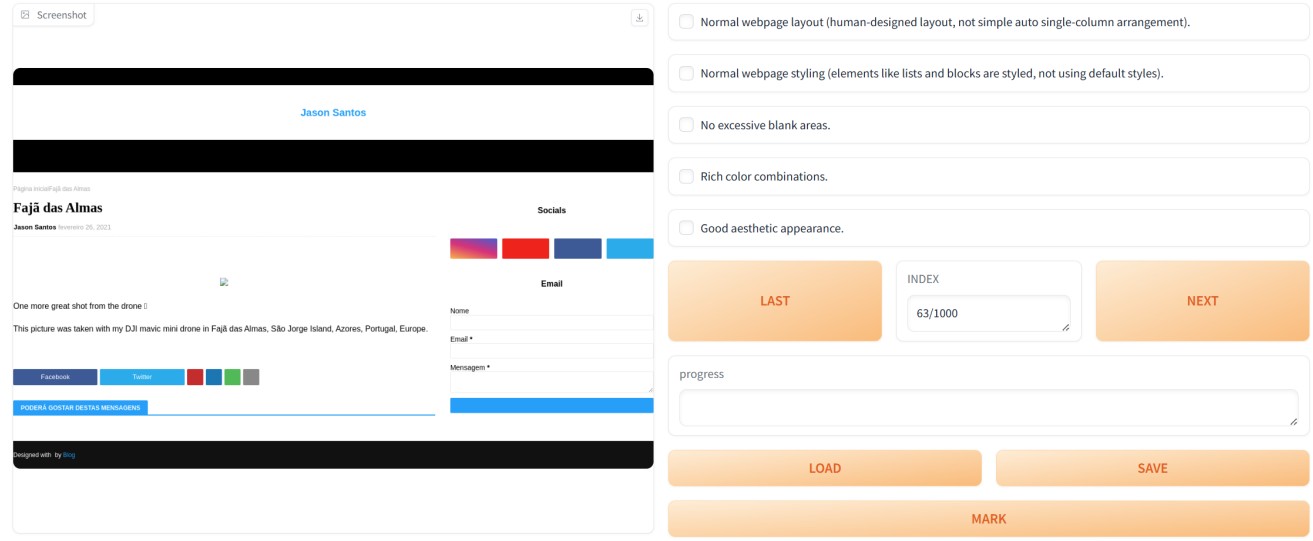

**Figure 10: The manual scoring tool.**

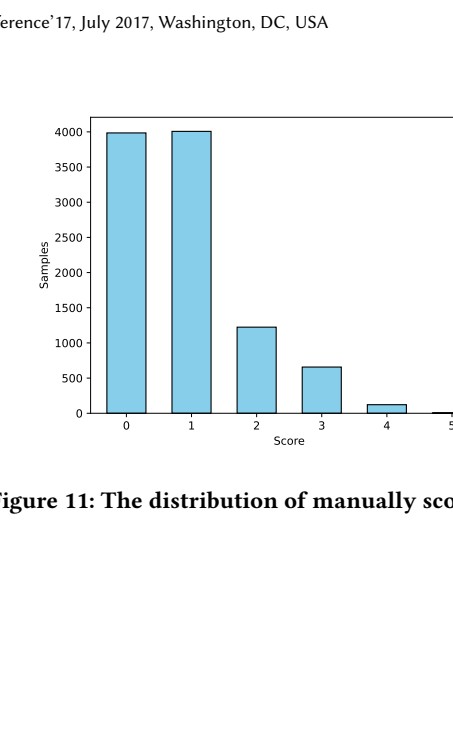

Figure 11: The distribution of manually scored results.

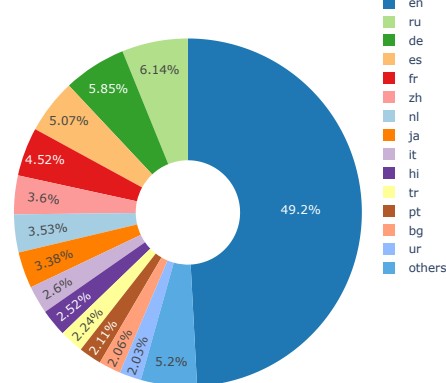

Figure 12: Language distribution in the training dataset.

## H Our Dataset on HuggingFace

We present a screenshot of our dataset uploaded to HuggingFace in Figure 14 and Figure 15. Detailed data cards and specifications for the dataset are available on HuggingFace.

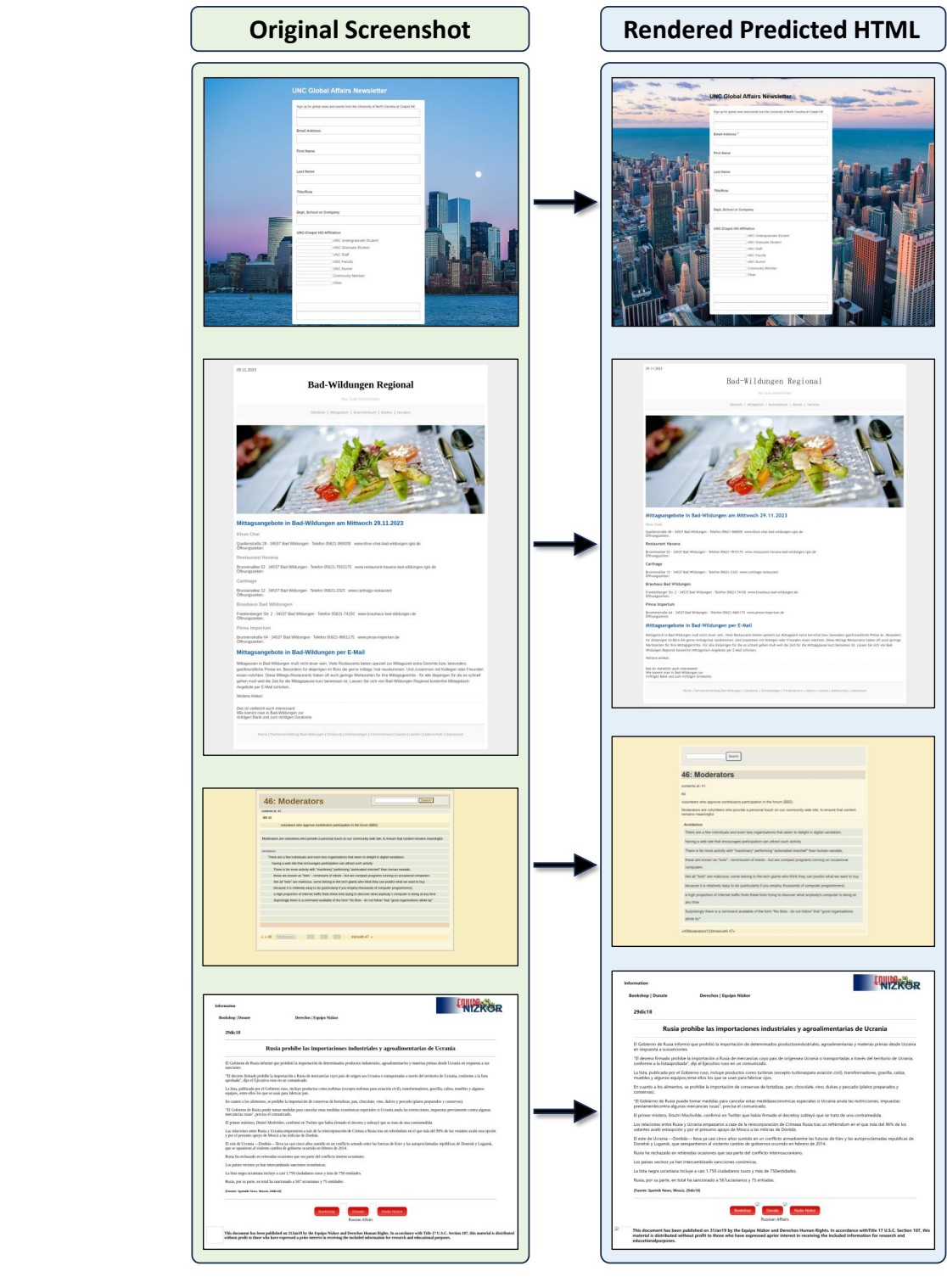

**Figure 13: Comparison of an original webpage (input) on the left, and the rendering of the HTML generated by our model, WebCoder, (output) on the right.**

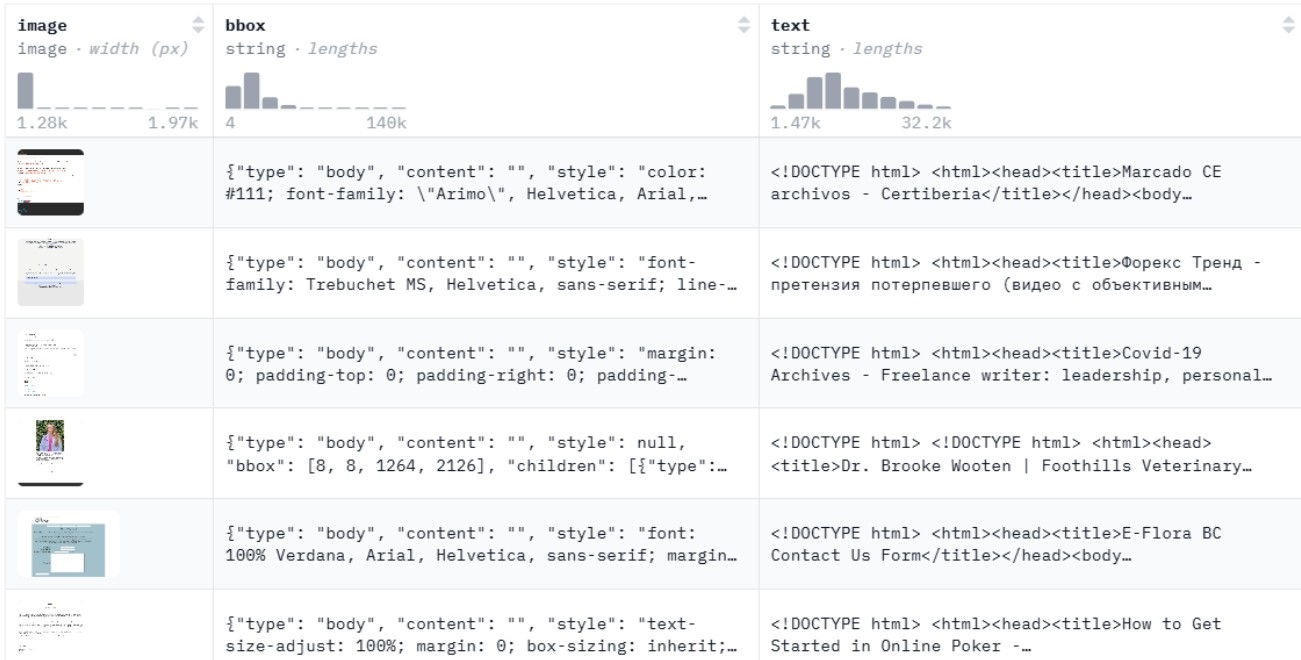

**Figure 14: Examples of WebCode2M dataset uploaded to HuggingFace (partial columns)**

| score int64 | scale sequence · lengths | lang string · classes | tokens sequence · lengths | hash string · lengths |
|---|---|---|---|---|
| 3 | [ 1280, 1561 ] | es | [ 678, 3344 ] | 370d891e8aa1b9eb1d935a6a76bccbf21a7329f0f0a6b3c4d05c b9fc4687318e |
| 3 | [ 1280, 1914 ] | ru | [ 1742, 4326 ] | 141a6998fe0b9578f5b2fd482bc44bf400e5ee950775d5c8a3da 10e1ff406500 |
| 3 | [ 1280, 1705 ] | en | [ 4081, 3733 ] | 6ba400fc0d1912d8ce2b9304d769329493c2b89e82ca518a00cd cfcc0fcc1cd2 |
| 3 | [ 1302, 2142 ] | en | [ 4063, 4729 ] | 02bc15699aa9151927d94f2e05e0ae0d987f648e6a3fccd009df 626c563f061d |
| 3 | [ 1280, 830 ] | en | [ 199, 796 ] | e422705a884c0c73fdf977af39328a00f1b69dedee61296f0e1f a6848b0ea1bc |
| 3 | [ 1370, 1683 ] | en | [ 2144, 3474 ] | a35a1c4750f514ea87e23f08c213de2e0e9ae284645b6cfc2545 f1682498f813 |

**Figure 15: Examples of WebCode2M dataset uploaded to HuggingFace (partial columns)**

