# OpenReview forum: "WebCode2M: A Real-World Dataset for Code Generation from Webpage Designs"
_ACM.org/TheWebConf/2025/Conference — WWW 2025 Oral_

### Official Review · Reviewer_H4k6 · 2024-11-13

**Novelty:** 4
**Technical Quality:** 6

**Review:**

This paper introduces WebCode2M, a new dataset comprising 2.56 million instances, each containing a design image along with the corresponding webpage code and layout details. The pipeline includes code purification, HTML rendering, filtering with neural scorer, and layout tree extraction. Then, the authors fine-tune a ViT model as a new baseline for translating webpage design images into HTML/CSS code. The main contribution is constructing a new dataset based on the Common Crawl dataset.

Weakness:
1. GPT-4o can perform this task relatively well. As a result, I'm uncertain about how significant this dataset is within the field.
2. I suspect that Table 3 may not fully demonstrate the effectiveness of the training set. It also seems possible that the better performance is due to the training and test sets being from the same domain.
3. This paper is a dataset paper and the authors propose a filtering strategy, but it is difficult to judge the quality of the dataset.
4. TreeBLEU has been used in previous paper [1], so TreeBLEU is not proposed in this paper.

[1] VISION2UI: A Real-World Dataset with Layout for Code Generation from UI Designs

**Questions:**

1. The dataset is based on the Common Crawl dataset. What is the quality of this dataset?
2. Why do all the baselines perform poorly on TreeBLEU? Please emphasize the importance of this metric.

**Reviewer Confidence:**

3: The reviewer is confident but not certain that the evaluation is correct

**Scope:**

2: The connection to the Web is incidental, e.g., use of Web data or API

---

### Official Review · Reviewer_jvBD · 2024-11-20

**Novelty:** 6
**Technical Quality:** 5

**Review:**

Strength
1. The authors propose the WebCode2M, which is a large-scale dataset for improving the capacity of generating webpage code from highfidelity images.
2. The authors finetune a ViT on the proposed dataset, and demonstrate the effectiveness of the dataset. Additionally, they propose a sub-tree based metric TreeBLEU to measure the structural hierarchy recall.

Weakness
1. The calculation of some mathematical quantities are not clear, such as the calculation of set S(t) in the metric TreeBLEU.
2. It would be better to detail the process of training ViT using the proposed dataset.

**Questions:**

1. What exactly does 1-height sub-tree refer to in the evaluation metric, and how to divide the generated code into the sub-trees of S(t)?
2. The trend of the TreeBLEU metric in Table 4 differs from the other two metrics. Is this because the metric that considers the sub-tree structure in the code reflects different aspects of the model compared to the other metrics?

**Reviewer Confidence:**

3: The reviewer is confident but not certain that the evaluation is correct

**Scope:**

3: The work is somewhat relevant to the Web and to the track, and is of narrow interest to a sub-community

---

### Official Review · Reviewer_osCd · 2024-11-27

**Novelty:** 5
**Technical Quality:** 5

**Review:**

**Summary**

This paper introduces WebCode2M, a large-scale dataset containing 2.56 million pairs of webpage designs and corresponding HTML/CSS code. The dataset aims to address limitations in existing datasets for automated webpage code generation, particularly the lack of high-quality, real-world training data. The authors also present a baseline model called WebCoder and introduce a new evaluation metric called TreeBLEU.

**Strengths**
1. The authors contributed a novel dataset (WebCode2M) that is large-scale, real-world, and quality-controlled.
2. The introduction of TreeBLEU metric for evaluating structural hierarchy recall
3. An analysis section for demonstrating the efficacy of the dataset via training a VLM

**Weakness**
I have one major concern left after reading the paper. I am not entirely convinced on the statement made in section 3.5. My concern is showing a VLM trained with WebCode2M performed better on WebCode2M does not actually prove its effectiveness. To make that point, one might need to evaluate the VLM trained with WebCode2M on other benchmarks and demonstrate superior performance.

**Questions:**

All in all, I think this is a solid research piece that will make good contribution to the community. Please see my raised weakness above.

**Reviewer Confidence:**

3: The reviewer is confident but not certain that the evaluation is correct

**Scope:**

4: The work is relevant to the Web and to the track, and is of broad interest to the community

---

### Official Review · Reviewer_UmeU · 2024-11-30

**Novelty:** 4
**Technical Quality:** 4

**Review:**

1. The introduction of WebCode2M, a large-scale dataset with 2.56 million high-quality instances, bridges the gap in multimodal datasets for webpage code generation. It includes design images, webpage code, and layout details, sourced from real-world scenarios, addressing a critical need in this domain.
2. The paper introduces TreeBLEU, a novel metric for assessing structural hierarchy recall, specifically tailored to webpage code generation tasks. Additionally, the baseline model, WebCoder, built on ViT, establishes a clear benchmark for fair comparisons and future research.｜
3. The dataset’s quality is ensured through a scoring model that filters instances based on aesthetic and completeness criteria. Combined with a focus on real-world applications, the dataset and method significantly enhance the performance of webpage code generation models, providing practical value for automating front-end development.

**Questions:**

1. It may be helpful to address instances of widow words in the text to improve the overall flow and readability of the manuscript.
2. Some images in the dataset comparison section appear slightly unclear. Enhancing their resolution could provide better visual clarity and support the comparison more effectively.
3. Could you clarify whether the subtree matching rules for TreeBLEU take into account specific HTML characteristics, such as the impact of semantic tags (e.g., `<header>`, `<article>`)? Considering these nuances might enhance the metric’s representation of HTML-specific features.
4.  Could you include some specific examples to demonstrate how TreeBLEU reflects the alignment of HTML DOM structures? This might help readers better understand its practical application and relevance.

**Reviewer Confidence:**

2: The reviewer is willing to defend the evaluation, but it is likely that the reviewer did not understand parts of the paper

**Scope:**

4: The work is relevant to the Web and to the track, and is of broad interest to the community